# Metaboloepigenetics: Role in the Regulation of Flow-Mediated Endothelial (Dys)Function and Atherosclerosis

**DOI:** 10.3390/cells14050378

**Published:** 2025-03-05

**Authors:** Francisco Santos, Hashum Sum, Denise Cheuk Lee Yan, Alison C. Brewer

**Affiliations:** 1School of Cardiovascular and Metabolic Medicine & Sciences, British Heart Foundation Centre of Research Excellence, Faculty of Life Sciences & Medicine, King’s College London, London SE5 9NU, UK; francisco.santos@kcl.ac.uk (F.S.); hashum.sum@kcl.ac.uk (H.S.); 2Faculty of Life Sciences & Medicine, King’s College London, London SE5 9NU, UK; denise.cl.yan@kcl.ac.uk

**Keywords:** endothelial dysfunction, atherosclerosis, shear stress, mechanotransduction, epigenetics, metabolism, metaboloepigenetics

## Abstract

Endothelial dysfunction is the main initiating factor in atherosclerosis. Through mechanotransduction, shear stress regulates endothelial cell function in both homeostatic and diseased states. Accumulating evidence reveals that epigenetic changes play critical roles in the etiology of cardiovascular diseases, including atherosclerosis. The metabolic regulation of epigenetics has emerged as an important factor in the control of gene expression in diseased states, but to the best of our knowledge, this connection remains largely unexplored in endothelial dysfunction and atherosclerosis. In this review, we (1) summarize how shear stress (or flow) regulates endothelial (dys)function; (2) explore the epigenetic alterations that occur in the endothelium in response to disturbed flow; (3) review endothelial cell metabolism under different shear stress conditions; and (4) suggest mechanisms which may link this altered metabolism to the regulation of the endothelial epigenome by modulations in metabolite availability. We believe that metabolic regulation plays an important role in endothelial epigenetic reprogramming and could pave the way for novel metabolism-based therapeutic strategies.

## 1. Introduction

Atherosclerosis is a cardiovascular disease characterized by chronic inflammation and thickened arterial walls due to lipoprotein-rich plaques, which form on the vascular endothelium, causing the narrowing of arterial lumen and the restriction of blood flow [1,2]. Resulting cardiovascular complications, such as ischaemic heart disease and strokes, are the leading causes of death worldwide [3]. Much of the pathophysiology of atherosclerosis is well defined, but, despite increased efforts in disease prevention, the global prevalence of cardiovascular diseases continues to grow [3,4].

Given its direct and constant contact with blood flow, the endothelium is regulated by haemodynamic forces induced by fluid shear stress, which is defined as the frictional force per unit area from flowing blood. Shear stress acts on mechanical sensing receptors (known as mechanosensors or mechanoreceptors) on endothelial cells (ECs), which transduce these mechanical cues into biochemical signals, a process called mechanotransduction, regulating EC physiology in both homeostatic and diseased states [5]. Examples of mechanosensors include membrane structures (cadherin, integrin, PECAM-1 and PlexinD1), receptors (G protein-coupled) and ion channels (Piezo1) (reviewed in [6]). Signalling pathways and critical nodes that are induced by shear stress and mechanotransduction include mitogen-activated protein kinases (MAPKs), phosphatidylinositol 3-kinase (PI3K)-AKT, mammalian target of rapamycin (mTOR), RHO GTPases and Yes-associated protein (YAP) and transcriptional coactivator with PDZ-binding motif (TAZ). Several of these pathways lead to the activation of shear stress-responsive transcription factors with well-known functions in EC biology, such as Krüppel-like factor 2 (KLF2) and KLF4 and nuclear factor erythroid 2-like (NRF2), in addition to hypoxia-inducible factor 1-alpha (HIF-1α) and nuclear factor kappa-light-chain-enhancer of activated B cells (NF-κB) [6].

The importance of the endothelium in the etiology of atherosclerosis cannot be disputed, and indeed, endothelial dysfunction is widely accepted as the initiating factor in the development of atherosclerosis (reviewed in [7]). This is underscored by the fact that early atherosclerotic lesions preferentially develop at branching or curved vascular regions, such as the carotid bifurcation in the lateral wall of the internal carotid artery, the proximal portion of the left anterior descending coronary artery and the lesser curvature of the aortic arch [8,9]. At these sites, disturbed flow predominates. This is detected by ECs, which subsequently elicit cellular responses that culminate in the induction of a pro-inflammatory phenotype and the initiation of atherosclerosis [7].

The dysregulation of the epigenome has been increasingly associated with the initiation and progression of atherosclerosis, which has been reclassified as an epigenetic disease (reviewed in [10,11,12,13]). Indeed, shear stress has been shown to regulate the epigenomic and transcriptomic signatures, as well as the metabolic profile, of ECs [7,14,15]. Epigenetic modifications are determined by both the abundance of epigenetic-modifying enzymes [16] and the availability of substrates, cofactors and donors that ultimately regulate the rate of these reactions. Thus, owing to their intrinsic kinetic and thermodynamic parameters, epigenetic-modifying enzymes are highly sensitive to fluctuations in the availability of certain metabolites caused by alterations in metabolic pathways, which consequently act to modulate the levels of specific epigenetic modifications [17]. Metaboloepigenetics, defined as the link between metabolism and the epigenetic control of gene expression, has been demonstrated to be crucial in cancer biology, immune cell activation and cell fate determination, as fluctuations in metabolites influence epigenetic alterations that contribute to diseased states and differentiation programmes [18,19,20]. Importantly, the endothelium has emerged as a metabolically dynamic organ that potentially regulates the epigenetic landscape in normal and dysfunctional ECs to effect important roles in vascular homeostasis and disease [21].

Here, we address the evidence that changes in shear stress link to the activation of specific mechanosensory pathways in ECs, which are associated with epigenetic regulation in both homeostasis and disease. We will focus on DNA methylation and histone post-translational modifications (PTMs). The metabolic reprogramming of dysfunctional ECs will also be discussed with regard to its impact on the availability of metabolites that can be used in epigenome-modifying reactions. To the best of our knowledge, a direct connection between epigenetic alterations and changes in metabolism in ECs has yet to be determined. Therefore, we speculate on the possible role of epigenetic regulation through metabolism in disturbed flow-induced endothelial dysfunction and atherosclerosis.

## 2. Flow-Mediated Endothelial (Dys)Function

The vascular endothelium is composed of a monolayer of ECs which lines the innermost layer of all blood vessels. ECs constitute the first barrier between blood and the rest of the vascular wall, regulating vascular permeability as a selective barrier that controls the passage of fluids, solutes and immune cells between blood and tissue [22]. In vivo, ECs are constantly exposed to blood flow, which regulates nearly every aspect of endothelial function. Stable blood flow, characterized by high-magnitude, unidirectional laminar shear stress, is observed in straight, non-branching regions of the vasculature and is associated with an atheroprotective phenotype. By contrast, branch points, curves and bifurcations are considered atheroprone regions. At these sites, rather than exerting unidirectional and pulsatile shear stress, laminar flow is disturbed and can be oscillatory and turbulent, thus exerting lower shear stress [23,24].

The effects of stable and disturbed flow have been studied in both in vivo and in vitro settings. Partial carotid ligation (PCL) has been extensively used in atheroprone mouse models (ApoE^−/−^ and Ldlr^−/−^) to induce and assess the effect of disturbed fluid shear stress in the development of atherosclerosis [25,26]. In this in vivo model, three caudal branches of the left common carotid artery (LCA) are surgically ligated without manipulating the LCA itself, therefore inducing disturbed flow in the LCA with characteristic low-magnitude oscillatory shear stress patterns. Importantly, the contralateral right carotid artery (RCA) continues to be exposed to stable flow [7]. In vitro flow models include the exposure of ECs to different shear stress conditions, either on a parallel-plate flow chamber or a microfluidic channel [7], thus mimicking the flow that ECs are exposed to in the vasculature.

KLF2 is an endothelial-expressed transcription factor that plays important roles in vascular homeostasis and is downregulated in ECs in dysfunctional states [27]. Stable flow has been shown to increase the transcription of KLF2 via the MEKK2/3–MEK5–ERK5 kinase cascade, which acts to downregulate the expression of pro-atherogenic genes [7,28]. KFL4 is another transcription factor that is essential for EC lineage [29]. Laminar (stable) shear stress leads to the upregulation of several atheroprotective genes via KLF4, such as *NOS3*, *THBD* and *ITPR3* [30,31] and has also been shown to regulate the integrity of adherens junctions by the phosphorylation and degradation of vascular endothelial (VE)-cadherin [32]. Laminar shear stress also promotes tight junction stability by regulating the expression of occludin and its attachment to the actin cytoskeleton [32,33,34]. Furthermore, atheroprotective shear stress has been shown to inhibit YAP/TAZ activity [35]. Functionally, laminar flow promotes the interaction between integrin β3 and Gα13, resulting in the inhibition of RhoA and phosphorylation (and subsequent inactivation) of YAP. Inactivation of YAP further supresses c-Jun NH2-terminal kinase (JNK) signalling, leading to reduced inflammation, thus promoting EC homeostasis [35].

One of the most important functions of the vascular endothelium is the production of nitric oxide (NO) from L-arginine by endothelial NO synthase (eNOS), which is encoded by the *NOS3* gene (Figure 1). NO acts on vascular smooth muscle cells (VSMCs) in the tunica media, thus regulating vascular tone by promoting smooth muscle relaxation and vasodilation [36]. Mechanistically, it has been found that stable flow induces eNOS activation via the activation of the PI3K-AKT pathway through a platelet endothelial cell adhesion molecule (PECAM)-1, VE-cadherin and vascular endothelial growth factor receptor (VEGFR)2/3 complex (Figure 1). NO also has anti-inflammatory effects, with potential roles in inhibiting NF-κB, decreasing the expression of monocyte chemoattractant protein-1 (MCP-1) and vascular adhesion molecule-1 (VCAM-1), thereby limiting leukocyte adhesion and safeguarding endothelial homeostasis (Figure 1) [36].

Changes in haemodynamic shear stress are a major cause for endothelium dysfunction and atherosclerotic lesions, which, as stated above, develop preferentially at vascular branches and curvatures, where disturbed flow predominates [7]. Disturbed flow has been shown to cause eNOS uncoupling and decrease the bioavailability of NO (Figure 1). Additionally, these atheroprone regions are also characterized by the upregulation of ICAM-1 and VCAM-1, leading to enhanced leukocyte adhesion and subsequent extravasation. Disturbed flow also leads to the internalization of VE-cadherin, increasing vascular permeability, and to the secretion of chemokines and cytokines, further enhancing inflammation (Figure 1) [37,38]. Of note, low shear stress has been shown to upregulate CXC chemokine-related transcripts, such as *CXCL12*, in a heparan sulfate proteoglycan-integrin β3-FAK-dependent manner. The activation of this signalling axis was shown to lead to the phosphorylation and activation of MAPKs p38β and p38δ, resulting in the activation of NF-Κb [39]. Integrin α5 has also been shown to be activated upon disturbed flow via the translocation of membrane lipid rafts. In Ldlr^−/−^ mice, the activation of integrin α5 was localized exclusively to atheroprone regions of the vasculature, which were associated with a pro-inflammatory phenotype [40]. The boosted EC inflammatory phenotype leads to enhanced leukocyte recruitment and adhesion and increased paracellular permeability. LDL may accumulate in areas where paracellular permeability has been compromised, known as paracellular pores, or may even be transported across ECs at atheroprone sites [41,42]. Importantly, although endothelial dysfunction and atherosclerosis develop preferentially at curved and branched regions, which are ubiquitous to every individual, the presence of other risk factors, such as hyperglycaemia, hypercholesterolaemia, obesity and smoking, potentiate endothelial dysfunction at these disturbed flow-exposed sites, thus increasing the risk of atherosclerosis [43].

## 3. Epigenetics and Regulation by Metabolism

Increasing evidence suggests that epigenetic alterations are important factors in the development of endothelial dysfunction and atherosclerosis [12]. Epigenetics encompasses the study of mechanisms which regulate gene expression, independent of the primary genetic sequence, by modulating chromatin structure and accessibility of *cis*-regulatory regions to the transcriptional machinery. Critical epigenetic mechanisms commonly involve the enzymatic addition and removal of covalent modifications to both DNA and histones of metabolite-derived groups [19].

Methylation is the main epigenetic modification on genomic DNA, and it typically occurs on cytosine (C)-rich motifs, known as CpG islands, which are usually located at or near gene promoters, and is typically associated with the repression of gene transcription [17]. Histones are subject to numerous PTMs involving metabolites, including acetylation, methylation, lactylation, phosphorylation and oxidation [17,44]. However, acetylation and methylation are the two most abundant and well-characterized PTMs. Respectively, histone-modifying enzymes utilize acetyl-CoA, S-adenosylmethionine (SAM), lactate, ATP and reactive oxygen species (ROS) as substrates (reviewed in [17,19,44]). These modifications act in specific and diverse ways to either increase or decrease the affinity of the histones to DNA (and hence affect the compactness of the chromatin) and can thus act as activators or repressors of gene expression [17,19].

The covalent modifications are catalyzed by enzymes, termed epigenetic writers and erasers, which add or remove epigenetic marks using metabolites as substrates, donors or cofactors [17]. DNA methylation reactions are carried out by DNA methyltransferases (DNMTs) using SAM as the methyl donor. DNMT1 is responsible for “copying” the original methylation signature of DNA during replication in which the newly synthesized strand (lacking methylation) is methylated, allowing the conservation of the DNA methylation profile in dividing cells [45], while DNMT3A and DNMT3B are de novo methyltransferases that introduce new methylation patterns that can be inherited by daughter cells [45]. Similarly, histone methylation reactions are carried out by histone methyltransferases (HMTs), which transfer methyl groups from SAM to arginine and lysine residues on histones H3 and H4 [19]. The methylation of arginine residues is associated with active transcription, while the methylation of lysine residues is linked to both active and repressed transcription depending on the residue that is methylated and the number of methyl groups that are added [46]. For instance, trimethylation on lysine (K) 4 of histone H3 (H3K4me3) is associated with active transcription, while trimethylation on K9 or K27 of histone H3 (H3K9me3 and H3K27me3, respectively) are repressive marks [17].

Thus, in both DNA and histone methylation reactions, the methyl group is derived from SAM, which is synthesized from methionine and ATP and serves as the universal methyl donor in numerous biological methylation processes [17]. Methionine uptake and one-carbon metabolism, which involves the inter-related folate and methionine cycles, therefore play an important role in methylation reactions. Upon the transfer of its methyl group, SAM is converted into S-adenosylhomocysteine (SAH), which can inhibit DNMTs and HMTs if accumulated [19]. Glycine, serine and threonine provide one-carbon units to tetrahydrofolate to transfer to homocysteine and facilitate the generation of SAM (Figure 2) [20].

The removal of DNA methylation can occur passively (at DNA replication or in the absence of DNMT-mediated “maintenance” of methylation) or by active mechanisms which occur independently of cell division. Notably, the ten-eleven translocation (TET) family of proteins (TET1-3) act as such epigenetic erasers by catalyzing the stepwise oxidation of 5-methylcysteine (5mC) to 5-hydroxymethylcytosine (5hmC), 5-formylcytosine (5fC) and 5-carboxycytosine (5aC) [17]. Unmethylated cytosine is restored by the excision of 5fC or 5caC by thymine-DNA glycosylase and base excision repair mechanisms [47]. TET enzymes are members of the 2-oxoglutarate-dependent dioxygenase (2-OGDD) superfamily, which require both molecular oxygen (O2) and α-ketoglutarate (α-KG) as substrates [17,19]. α-KG is a key intermediate in cellular metabolism, primarily produced in the TCA cycle within the mitochondria, through the conversion of isocitrate by isocitrate dehydrogenase (IDH) or by the anaplerotic glutaminolysis pathway [18]. Furthermore, due to their structural similarity to α-KG, the metabolites immediately downstream of α-KG in the TCA cycle (succinate and fumarate) and 2-hydroxyglutarate (2-HG), which can be generated by dysregulated/mutant IDH, can act as inhibitors of TET demethylases [19]. Thus, both the abundance of α-KG and the balance between α-KG and its antagonists may impact the activities of TET enzymes. Hence, the rates of DNA demethylation may, like those of methylation, be impacted by cellular metabolism [19]. Similarly, the demethylation of histones can be carried out by Jumonji-domain-containing histone demethylases (JHDMs) which, like TETs, are members of the 2-OGDD superfamily and require O2 and α-KG [17,19]

While histone methylation is associated with both active and repressive marks, histone acetylation is exclusively associated with permissive gene transcription [20]. Upon the transfer of acetyl groups, by reactions catalyzed by histone acetyltransferases (HATs), histones become less bound to negatively charged DNA, promoting a more open chromatin structure [48]. Histone acetylation is deeply impacted by the availability of acetyl-CoA, which is the main donor of acetyl groups for acetylation reactions and can be generated through several metabolic pathways, including glycolysis, through the conversion of pyruvate into acetyl-CoA by pyruvate dehydrogenase, fatty acid oxidation (FAO) and de novo lipogenesis [48,49]. Because acetyl-CoA cannot diffuse across membranes, it condenses with oxaloacetate to produce citrate, which is mobilized to the cytoplasm through the malate–citrate antiporter system. In the cytoplasm, citrate is then converted into acetyl-CoA by ATP citrate lyase (ACLY), which plays an important role in maintaining the levels of acetyl-CoA for histone acetylation [48]. Acetyl-CoA can also be generated from acetate by acetyl-CoA synthetase 2 (ACSS2). Both ACLY and ACSS2 are found in the cytoplasm but can also be present in the nucleus, where they are believed to produce high levels of acetyl-CoA, likely near sites of histone acetylation [20]. The removal of histone acetylation marks is catalyzed by histone deacetylases (HDACs). Class I, II and IV HDACs are influenced by the availability of butyrate and β-hydroxybutyrate, produced during FAO and ketogenesis [20]. These ketone bodies are inhibitors of HDACs and are therefore associated with an open chromatin conformation and active gene expression. Sirtuins (SIRTs; class III HDACs) use oxidized nicotinamide adenine dinucleotide (NAD+) as a cofactor, providing a link between levels of NAD+ and SIRT-mediated histone deacetylation [50]. Thus, these covalent epigenetic marks are impacted by both the expression of epigenetic writers and the availability of metabolite co-factors and hence metabolic pathways [19].

## 4. Epigenetic Alterations Mediated by Disturbed Flow

### 4.1. DNA (De)Methylation

Epigenetic modifications, including DNA modifications, are recognized to be highly dynamic and to be impacted by environmental stimuli [51]. In cardiovascular diseases, the association of aberrant DNA methylation with atherosclerosis development is increasingly being studied. At a global level, hypomethylation within atherosclerotic tissue has typically been observed, while locus-specific aberrant hypermethylation on promoter regions of specific genes known to be involved in disease pathology are common hallmarks in atherosclerosis (reviewed in [11]). Although technically challenging, several studies have also addressed and reported changes to methylation patterns, specifically with the endothelium of atherosclerotic tissue [11].

The effects of both physiological (high-magnitude and unidirectional) and pathological (low-magnitude or disturbed) flow upon the methylome and transcriptome of underlying endothelium has also been investigated in several studies both in vivo and in vitro. Disturbed fluid shear stress has been shown to dysregulate the expression of DNMTs and TET enzymes in ECs, impacting the endothelial methylome [15,52]. In vivo, the aortic EC methylomes in sites of disturbed (aortic arch) and physiological, unidirectional (descending aorta) flow were compared in pigs. In this study, genome-wide methylated DNA immunoprecipitation sequencing (MeDIP-seq) identified over 5500 differentially methylated regions (DMRs), predominantly in exons and 5’UTRs of annotated genes, of which 60 were linked to cardiovascular disease [24,53]. Differential methylation in selected genes, including *HOXA* genes and *ATF4*, was further confirmed by methylation-specific PCR. In mice subjected to PCL, an increase in DNMT1 was observed under disturbed flow compared to the artery used as an undisturbed flow control [24]. Furthermore, the application of PCL in ApoE^−/−^ mice led to the discovery that regions where flow was disturbed were significantly more susceptible to the development of atherosclerosis. Strikingly, treatment with the DNMT inhibitor 5-aza-2′-deoxycytidine (5-Aza) reversed these effects. Using reduced representation bisulfite sequencing, the authors demonstrated genome-wide DMRs within the endothelial-enriched genomic DNA in the LCAs compared to the RCAs, highlighting the importance of DNA methylation in endothelial dysfunction [24]. Furthermore, in combination with microarray transcriptomic analyses, 11 “mechanosensitive” genes were identified in this study whose transcriptional expression was regulated by DNA methylation. Many of the 11 genes identified are known to be important regulators of endothelial cell function and homeostasis and/or are functionally associated with atherosclerosis. Several more flow-responsive genes have also been identified in subsequent studies. These are summarized in Table 1.

In vitro data collected using human umbilical vein ECs (HUVECs) are consistent with these in vivo results. HUVECs subjected to disturbed flow also presented higher levels of DNMT1, which were associated with enhanced monocyte adhesion [24]. The disturbed flow-induced upregulation of DNMT1 has been linked to increased activity of mTOR, thus providing a link between mechanotransduction and the regulation of epigenetic-modifying enzymes. Oscillatory shear stress was found to activate the PI3K-mTOR-p70S6K through integrin β3, resulting in the upregulation of DNMT1. Importantly, disturbed shear stress coupled with the pharmacological inhibition of this signalling pathway resulted in decreased expression levels of DNMT1 and an improvement in endothelial function [90]. Upon exposure of human aortic ECs to disturbed shear stress, DNMT3A was also found to be upregulated at the protein level, resulting in the hypermethylation of the *KLF4* gene promoter and leading to the repression of KLF4 target genes, such as *NOS3* [91]. Consistent with this, increased methylation levels on the *KLF4* promoter in ECs from swine aortic arch were also observed [53,91].

The hypermethylated state and associated endothelial dysfunction observed in ECs, upon exposure to disturbed flow, may also be attributed to the downregulation of TET enzymes, particularly TET2 [52,92]. Thus, HUVECs exposed to low shear stress in vitro were characterized by lower expression levels of TET2 compared to HUVECs exposed to physiological shear stress. The downregulation of TET2 was accompanied with a decrease in autophagic flux, which was demonstrated by an increase in p62, decreased levels of LC3 lipidation and decreased protein levels of eNOS [52]. Importantly, the overexpression of TET2 before low shear stress exposure recovered eNOS and ameliorated autophagy. Low shear stress-mediated downregulation of TET2 has also been associated with enhanced endothelial–mesenchymal transition (EndMT) [92]. In HUVECs, the decrease in TET2 was associated with a reduced expression of endothelial marker VE-cadherin and an increase in mesenchymal marker vimentin. The knockdown of TET2 revealed an increase in the protein levels of vimentin, alpha smooth muscle actin (α-SMA) and fibroblast-specific protein 1 (FSP-1), which were associated with a rearrangement of the cytoskeleton and increased cell migration. Despite the importance of these studies on elucidating the role of low shear stress on the regulation of TET2, it is unclear whether the reported effects are dependent on the impairment of hydroxymethylation, leading to DNA hypermethylation. However, decreased levels of TET2 and 5hmC were shown in aortic sinuses and atherosclerotic lesions of ApoE^−/−^ mice, while 5mC increased, when compared to normal vascular tissue [52,93]. Importantly, the overexpression of TET2 in ApoE^−/−^ mice subjected to PCL inhibited the development of atherosclerotic lesions, even preventing the accumulation of lipids [92]. TET2 overexpression was also shown to decrease the expression of pro-inflammatory molecules involved in endothelial dysfunction, such as ICAM-1, VCAM-1 and MCP-1 [93]. Together, this shows that TET2 potentially protects against atherosclerosis, but its catalytic role in the regulation of DNA methylation in the setting of disturbed flow remains to be confirmed.

### 4.2. Histone PTMs

Several studies have highlighted the importance of histone modifications as regulators of endothelial-specific gene expression. For instance, HAT7 has been implicated in the acetylation of histones H3 and H4, activating the expression of vascular endothelial growth factor receptor and maintaining EC identity [94]. Specifically in ECs, the endothelial-expressed *NOS3* promoter exhibits the enrichment of active histone marks, such as H3K9ac, H4K12ac, H3K4me2 and H3K4me3 [95]. Strikingly, in HUVECs, hypoxia leads to the loss of these active histone marks from the *NOS3* proximal promoter, highlighting their dynamic regulation [96].

Histone PTMs have also emerged as important regulators of EC-specific gene expression in the setting of altered blood flow. In a seminal study, Illi and colleagues found that HUVECs exposed to shear stress led to the increased acetylation and phosphorylation of histone H3 (H3K14ac and H3S10ph, respectively) [97]. Furthermore, shear stress coupled with treatment with HDAC inhibitor trichostatin A also led to the acetylation of histone H4 [97]. Increased activity of HAT p300 was also observed in HUVECs exposed to laminar shear stress [98]. This is associated with increased acetylation levels of histones H3 and H4 at the shear stress response element (SSRE) site within the *NOS3* promoter, resulting in increased chromatin accessibility and transcriptional activity. Importantly, the inhibition of p300 by curcumin prevented the shear stress-induced transcription of *NOS3* [98]. In vivo, an increase in H3K9ac was observed in mouse descending aorta, where stable flow predominates [99].

Several reports have suggested that shear stress also regulates the expression and/or activity of HDACs. Immunohistochemical staining of ApoE^−/−^ mouse ECs isolated from arterial branches (where disturbed flow is prevalent) revealed an increase in HDAC3, which was confirmed in HUVECs exposed to low shear stress [100]. Moreover, further in vitro studies verified that disturbed shear stress also leads to the upregulation and nuclear accumulation of HDAC1, 2, 3, 5 and 7 [101]. Importantly, HDAC1 deacetylates the *NOS3* promoter, repressing its transcription [102]. Disturbed flow was also shown to promote the association of HDAC1, 2 and 3 to NRF2 protein, leading to its deacetylation and, consequently, the decreased expression of antioxidant genes [101]. SIRTs are also suggested to be regulated by flow. Consistent with this, the expression of SIRT1 was found to be elevated in the thoracic region of the mouse aorta compared to the aortic arch [103]. This was further confirmed in HUVECs exposed to laminar and oscillatory shear stress. In this study, the increase in SIRT1 was shown to be involved in the deacetylation of eNOS protein, enhancing its activity [103]. It is important to point out that some of these studies report the deacetylation of non-histone proteins; however, it is likely that changes in such epigenetic-modifying enzymes would also impact chromatin remodelling.

Histone methylation has also been demonstrated to be sensitive to laminar and oscillatory shear stress, with studies suggesting decreased and increased levels of H3K27me3 and H3K9me3, respectively, in HUVECs exposed to low shear stress [99]. Moreover, using a macro-channels model system, the exposure of HUVECs to disturbed flow caused a generalized increase in H3K4me3 and a decrease in H3K27me3 [104]. Treatment with TNF-α yielded similar results, suggesting that the gain and loss of H3K4me3 and H3K27me3, respectively, is linked to a dysfunctional phenotype in ECs [104]. Consistent with this, tissue isolated from the aortic arch of ApoE^−/−^ mice also showed a decrease in H3K27me3. Perhaps surprisingly, however, this was accompanied by an upregulation of EZH2, which is responsible for the deposition of this histone mark [105].

Recently, histone phosphorylation was shown to regulate disturbed flow-induced endothelial inflammation. Mechanistically, disturbed flow activates the integrin α5β1-PKN1 axis, inducing the phosphorylation of activator protein-1 (AP-1) transcription factor subunit JUN and its translocation into the nucleus, where it associates with FOS to promote the expression of pro-inflammatory genes [106]. PKN1 also translocates into the nucleus, where it phosphorylates histone H3.3 at serine 31 (H3.3S31ph), activating the expression of *FOS*/*FOSB* and thus enhancing endothelial inflammation [106]. Interestingly, histone phosphorylation has been shown to induce the activity of other histone-modifying enzymes and promote other histone PTMs, such as acetylation and methylation, suggesting that epigenetic alterations are not “compartmentalized” but rather highly collaborative [107,108].

## 5. A Possible Role for Metaboloepigenetics in Endothelial Dysfunction

As discussed above, disturbed flow can act to reprogram the epigenome of ECs, in part via the dysregulation of the expression and/or localisation of epigenetic-modifying enzymes. Importantly, however, the rates of the reactions catalyzed by these enzymes are also dependent on the availability of metabolites that act as substrates, cofactors and donors, therefore linking epigenetics to metabolic alterations [17]. It has become well established that metabolism and metabolic fluxes play an important role in endothelial dysfunction (reviewed in [14,21]). Most studies focused on EC metabolism have been limited to cultured ECs under static conditions, which do not consider the shear stress that ECs are constantly exposed to in vivo. Nonetheless, a few studies have reported alterations in metabolism and mitochondrial function upon different fluid shear stress conditions [41,109]. This section summarizes what is currently known about EC metabolism in both homeostatic (exposed to laminar flow) and disturbed flow-induced dysfunctional states and hypothesizes how metabolic changes may impact epigenetic alterations linked to EC (dys)function.

### 5.1. Shear Stress-Mediated EC Metabolism

Flow-mediated EC metabolism appears to differ significantly from that reported under static conditions (reviewed in [110]). HUVECs exposed to laminar shear stress are characterized by increased mitochondrial metabolism and remodelling of the mitochondrial network to comprise mostly more elongated, tubular mitochondria that arise from fusion events (i.e., when two mitochondria merge) [41,60]. Mitophagy was also reported to be more active in laminar shear stress-exposed HUVECs [111]. Indeed, the removal of damaged mitochondria by mitophagy creates an environment that favours mitochondrial metabolism [112]. The induction of mitophagy has been shown to be necessary for the endothelial differentiation of iPSCs, where PINK1-mediated mitophagy was shown to precede a boost in mitochondrial biogenesis, with increased expression of peroxisome proliferator gamma coactivator-1α (PGC-1α) [113]. Perhaps significantly, PGC-1α has recently emerged as a flow-responsive gene. Thus, compared with oscillatory shear stress, human aortic ECs exposed to undisturbed flow exhibited an increased expression of PGC-1α [60], consistent with an increase in mitochondrial biogenesis and function. Complex and mature mitochondrial networks are associated with cells that are more dependent on oxidative phosphorylation (OXPHOS). FAO, which has been shown to be important for EC identity (Figure 2) [114], generates reducing equivalents that enter OXPHOS [115], thus supporting mitochondrial metabolism. Consistently, the upregulation of KLF2 by laminar shear stress has been shown to decrease glucose uptake, thus limiting glycolysis [109]. These studies possibly suggest that, unlike what has been observed in cultured ECs in static conditions (reviewed in [21,110,116]), mitochondrial metabolism might play an important role in EC bioenergetics and homeostasis.

By contrast, enhanced glycolysis has been reported to take place in ECs at regions of disturbed flow (Figure 2) [117]; however, it is still unclear as to whether this is an adaptive, atheroprotective response or increases the susceptibility to atherosclerosis [117]. Low shear stress leads to the activation of mechanotransducers YAP/TAZ, which promote glycolysis and a pro-inflammatory phenotype [118]. In an independent study, ECs isolated from atheroprone regions of porcine aorta exhibited higher levels of HIF-1α, leading to the activation of several glycolytic genes, including 6-phosphofructo-2-kinase/fructose-2,6-biphosphatase 3 (*PFKFB3*), hexokinase 2 (*HK2*), enolase 2 (*ENO2*) and glucose transporters 1 and 3 (GLUT1/3) [72]. Mechanistically, disturbed flow led to an increase in NADPH oxidase 4-generated ROS, which were shown to be responsible for the increased stabilization of HIF-1α [119]. In human aortic ECs exposed to disturbed flow, an increase in HIF-1α was also observed, which was accompanied with enhanced glycolysis and a decrease in mitochondrial respiratory capacity (Figure 2) [119]. Strikingly, lactate dehydrogenase A (*LDHA*) was found to be upregulated upon disturbed flow, possibly limiting the availability of pyruvate to enter the TCA cycle and reducing OXPHOS [119]. Consistently, human aortic ECs exposed to disturbed flow present defective mitophagy, a very fragmented mitochondrial network and an increase in mitochondrial ROS compared to ECs exposed to stable flow [41,111]. Fragmented mitochondrial networks, composed of round and unfused mitochondria, are usually present in cells that mainly rely on glycolysis for ATP production, which is consistent with the HIF-1α-driven increase in aerobic glycolysis observed in ECs exposed to low shear stress. Disturbed flow also leads to an increase in enolase1 (*ENO1*), further potentiating the glycolytic pathway (Figure 2) [41]. The phenotype caused by disturbed flow was shown to be pro-inflammatory, with an increased engagement in TGF-β signalling and EndMT [41]. Accordingly, ECs treated with TGF-β, a major activator of EndMT, were characterized by a decrease in FAO [114]. Recently, EPAS1 (endothelial PAS domain-containing protein 1, also known as *HIF2A*) was shown to protect ECs in regions of disturbed flow by upregulating fatty acid transporter cluster of differentiation 36 (*CD36*) and endothelial lipase (*LIPG*), promoting fatty acid uptake and oxidation, and thus maintaining EC proliferation. The authors further demonstrated that obesity causes a downregulation of EPAS1, leading to increased susceptibility to atherosclerotic lesions at sites of disturbed flow [120]. These studies further suggest that FAO possibly plays an important role in EC identity and homeostasis [120].

### 5.2. Linking the EC Epigenome and Metabolism

The epigenetic alterations that take place upon disturbed flow-induced endothelial dysfunction might suggest that metabolic reprogramming occurs, perhaps in part, to regulate the availability of relevant metabolites to fuel epigenome-modifying reactions. However, this connection between metabolic regulation and epigenetic alterations has not been fully investigated in the setting of endothelial (dys)function thus far. As stated above, studies have shown that ECs exposed to different types of shear stress (laminar/undisturbed vs. oscillatory/disturbed) are characterized by the preferential use of specific metabolic pathways, and ECs also exhibit distinct, shear stress-dependent mitochondrial networks. Many of the epigenetic-modifying metabolites are produced in the mitochondria, so changes in mitochondrial morphology and function will directly impact the availability of these metabolites [18,121,122].

#### 5.2.1. Acetyl-CoA and Histone Acetylation

FAO is a major source of acetyl-CoA that can be used in histone acetylation reactions [123] and has been shown to be important for EC identity, possibly by maintaining the intracellular pool of FAO-derived acetyl-CoA [114]. Moreover, as stated above, loss of EC identity upon the induction of EndMT was reported to be accompanied with a decrease in FAO, which resulted in lower levels of acetyl-CoA [114]. FAO, however, is not the only source of acetyl-CoA for histone acetylation, and fluctuations in the availability of differentially sourced acetyl-CoA are crucial for the regulation of histone acetylation and transcriptional regulation. For example, in iPSCs, the deprivation of exogenous lipids has been shown to cause an exacerbation of de novo lipogenesis via citrate, thus increasing the intracellular levels of lipogenic acetyl-CoA [49,115]. Mechanistically, it was shown that citrate diverges from the TCA cycle to the cytoplasm, where it is converted into acetyl-CoA by ACLY (Figure 2). Significantly, in the same study, enhancing FAO did not result in the same effect on the levels of acetyl-CoA and histone acetylation [49,115]. Also, in this setting, an increase in histone acetylation marks was associated with increased HAT7 and decreased SIRT1 and HDAC1 expression levels [49], suggesting that altered expression of epigenetic-modifying enzymes and metabolic reprogramming may be linked. Interestingly, oscillatory shear stress was shown to upregulate fatty acid synthase (*FASN*), possibly suggesting dysregulated de novo lipogenesis, but further studies are needed. In proliferating muscle stem cells, regeneration is promoted when glucose is not completely oxidized, also causing citrate to be diverged to the cytoplasm and be converted into acetyl-CoA [124]. Interestingly, upon the induction of EndMT with TGF-β, glucose-derived acetate was shown to increase the intracellular pool of acetyl-CoA, in an ACSS2-dependent manner (Figure 2), in human umbilical artery ECs [125]. This further emphasizes that acetyl-CoA derived from different sources may be used for the acetylation of histones in distinct genomic regions.

#### 5.2.2. SAM and DNA/Histone Methylation

Animal-derived proteins are known to have atherogenic effects, mainly due to their high methionine content. Indeed, excessive methionine intake contributes to endothelial dysfunction and atherosclerosis [126]. Because methionine is the main precursor to SAM, which donates its methyl group for DNA and histone methylation reactions (Figure 2) [17], one could argue that the increased utilization of methionine and/or the replenishment of carbon units could fuel the DNA hypermethylation observed in ECs when exposed to disturbed flow. Increased methionine uptake or the synthesis of amino acids that are involved in SAM metabolism have been observed in several types of cancer to alter the intracellular pool of SAM and methylation levels of DNA and histones [127,128]. Changing the intracellular pool of SAM can impact methylation patterns and thus influence cell fate decisions [17]. Perhaps significantly, one-carbon metabolism has been reported to be downregulated upon the exposure of human ECs to both laminar and oscillatory shear stress compared to static conditions [129]. Furthermore, methionine was shown to be increased in blood plasma samples obtained from ApoE^−/−^ mice subjected to PCL, possibly hinting at altered methionine uptake by ECs [130] (Figure 2). Thus, the effects of flow upon methionine uptake, one-carbon metabolism and methylation need to be further explored.

#### 5.2.3. α-KG and DNA/Histone Demethylation

α-KG is essential for the demethylation of DNA and histones, and fluctuations in this metabolite have been shown to influence demethylation levels and, consequently, cell fate [131]. For instance, mouse ESCs have been shown to metabolize both glucose and glutamine to preserve high levels of α-KG and promote histone and DNA demethylation to determine cell fate [131]. Recently, abrogating FAO in mouse cardiomyocytes through the depletion of *Cpt1b* (the gene that encodes the muscle-specific isoform of FAO rate-limiting enzyme carnitine palmitoyltransferase 1B) demonstrated an increase in the levels of α-KG associated with a reduction in the levels of H3K4me3 in cardiomyocyte-specific genes and enhanced cardiac regeneration [132]. The fragmented mitochondrial network that characterizes ECs exposed to oscillatory shear stress might suggest a dysregulation of the TCA cycle and altered mitochondrial metabolism [133], possibly impacting the production of α-KG by IDH. However, the pro-inflammatory YAP/TAZ pathway, activated upon endothelial dysfunction, also enhances glutaminolysis, which may also impact the levels of α-KG [21].

#### 5.2.4. Lactate and Histone Lactylation

Disturbed flow-exposed ECs have been shown to rely on aerobic glycolysis. Despite glycolysis also being a potential source of acetyl-CoA [134], the aerobic glycolysis observed upon disturbed shear stress might rather favour the generation of lactate. In dysfunctional ECs, *LDHA* is upregulated, promoting the conversion of pyruvate into lactate (Figure 2) [119]. Another important histone PTM, although less explored in EC biology, is lactylation. The roles of histone lactylation have been reviewed elsewhere [135], and increased levels of this PTM have been linked to an increase in glycolysis [136,137]. Recently, lactate derived from aerobic glycolysis in ECs was shown to induce histone lactylation on H3K18 (H3K18la) on the *SNAI1* promoter, activating its expression and promoting EndMT [138]. The inhibition of glycolysis decreased H3K18la levels on the *SNAI1* promoter and attenuated EndMT and atherosclerosis [138].

## 6. Conclusions and Future Perspectives

The observation that atherosclerosis develops preferentially at sites of disturbed flow highlights the critical role if mechanosensory pathways in the endothelium in the etiology of the disease. Understanding the molecular mechanisms which underlie the reprogramming and remodelling of the vasculature in response to mechanosensory cues in ECs is crucial to inform the development of better therapeutic or preventive strategies to target cardiovascular diseases. Current research methods, particularly those involving dynamic changes in epigenetic marks, have limitations. However, the continual development of more sensitive and specific assay systems may further promote future progress.

Epigenetic changes within ECs are increasingly demonstrated to be causally related to disease development and progression. Here, we explore how metabolic changes, induced by mechanosensory pathways, might act to change the epigenetic landscape through alterations in the availability in metabolites. The levels of acetyl-CoA, α-KG and SAM are all known to regulate chromatin remodelling. EC identity seems to be compromised when FAO is downregulated. However, to the best of our knowledge, whether the decrease in FAO is causal in the loss of histone acetylation at EC-relevant *loci* is unknown, thus warranting future studies. Acetyl-CoA derived from different metabolic pathways does not seem to exert the same effect on histone acetylation and cell identity. For instance, FAO- and lipogenic-derived acetyl-CoA do not have the same impact on maintaining histone acetylation in human iPSCs. Interestingly, the increase in *FASN* in oscillatory shear stress-exposed ECs may suggest an increase in de novo lipogenesis, but this also needs to be confirmed. Furthermore, the hypermethylation of EC-relevant gene promoters was found to take place upon exposure to disturbed flow, suggesting that DNA methylation reactions may be fueled by an increased uptake of methionine and subsequent conversion to SAM, but further studies are needed to assess this. Glucose-derived lactate has been demonstrated to fuel histone lactylation upon endothelial dysfunction, thus prompting the loss of EC identity. Nonetheless, flow-dependent increases in lactate and histone lactylation remain largely unexplored, and the role of this modification in dysfunctional ECs need to be further investigated. It should also be noted that the roles of altered metabolite levels to specifically modulate the epigenome cannot be assumed and remain to be demonstrated definitively in many cases.

The direct impact of metabolism on the EC epigenome remains largely uninvestigated. However, given their interconnection, we believe it is reasonable to expect that shear stress-induced metabolic shifts impact the EC epigenome, paving the way to target metabolism as a therapeutic strategy. Lessons from cancer metabolism could be of use when studying the relationship between metabolism and epigenetics in ECs. The altered metabolism observed in cancer has increasingly been targeted by pharmacological inhibition. Indeed, targeting glycolysis, FAO and OXPHOS has successfully been shown to compromise cancer cell metabolism and hamper disease aggressiveness and progression (reviewed in [139]). Significantly, targeting glycolysis has been shown to be a viable strategy to mitigate the effects of atherosclerosis. Indeed, the inhibition of HIF-1α and PFKFB3 has been shown to reduce atherosclerosis in a mouse model [140,141]. Also, increasing FAO by overexpressing PPAR-γ has shown to mitigate FA-induced endothelial dysfunction observed in obesity [110]. Targeting ACLY has become a therapeutic target to limit FA and cholesterol biosynthesis, both of which require acetyl-CoA in the cytoplasm [14].

Although disturbed flow, which by itself causes a significant degree of metabolic reprogramming, shifts ECs towards a dysfunctional state, and the effect of other risk factors on EC metabolism, such as hyperglycaemia, hypercholesterolaemia and obesity, should also be considered. Furthermore, despite the importance of ECs in maintaining vascular integrity, the contribution other cell types in the vasculature (VSMCs and immune cells) should not be ignored when addressing the pathogenesis of atherosclerosis [2]. Indeed, mutations in DNMTs and TETs that occur during the clonal haematopoiesis of indeterminate potential (CHIP) have been reported to potentiate inflammation in macrophages, increasing the risk of atherosclerosis.

Altogether, mechanotransduction is regulated by shear stress and is involved in the regulation of epigenetic-modifying enzymes, and we believe that metabolic alterations might fuel the epigenetic reprogramming that takes place upon exposure to disturbed flow, which is associated with endothelial dysfunction. However, intercellular communications need to be considered to provide a comprehensive overview of the development of endothelial dysfunction and atherosclerosis. 

## Figures and Tables

**Figure 1 cells-14-00378-f001:**
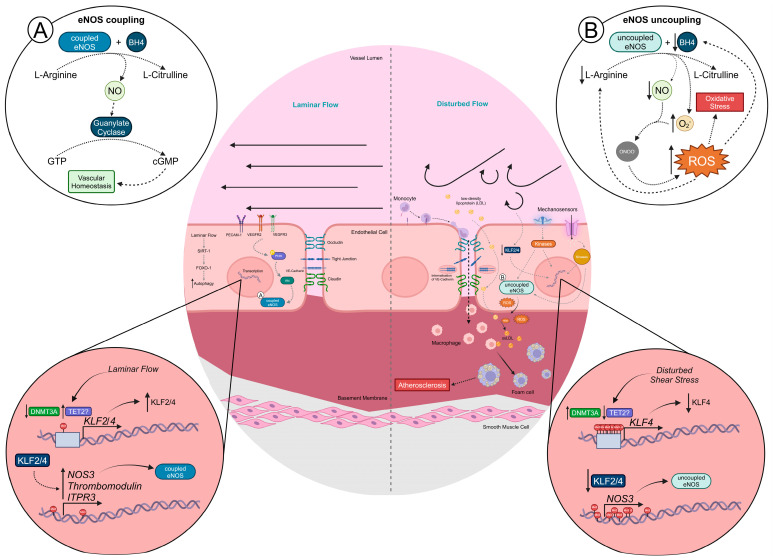
Endothelial dysfunction develops at sites of disturbed flow. (**A**) Atheroprotective laminar shear stress (stable flow) acts on EC mechanosensors, which then activate signalling pathways. This leads to the active transcription of genes that encode transcription factors known to be important for EC homeostasis and identity, such as KLF2/4. The production of NO from L-arginine, by eNOS, is an important feature of a healthy endothelium as it acts on vascular smooth muscle cells to promote vasodilation. (**B**) On the contrary, oscillatory shear stress (disturbed flow) leads to the activation of alternative signalling pathways that end up upregulating DNMTs, repressing the expression of KLF2/4. Disturbed flow-induced endothelial dysfunction causes an upregulation of cell adhesion molecules, leading to an exacerbation in leukocyte adhesion and transcellular migration. Furthermore, oxidative stress also increases upon disturbed flow, contributing to the oxidation of LDL into oxidized LDL, which is taken up by macrophages, causing them to reprogram into foam cells, which are an important feature in the development of atherosclerosis. Abbreviations: Akt (or PKB), protein kinase B; cGMP, cyclic guanosine monophosphate; DNMT, DNA methyltransferase; eNOS, endothelial nitric oxide synthase; FOXO-1, forkhead box protein O1; GTP, guanosine triphosphate; *ITPR3*, gene that encodes inositol 1,4,5-trisphosphate receptor, type 3; KLF, Krüppel-like factor; LDL, low-density lipoprotein; NO, nitric oxide; *NOS3*, gene that encodes eNOS; O_2_^−^, superoxide; ONOO^−^, peroxynitrite; oxLDL, oxidized low-density lipoprotein; PECAM, platelet endothelial cell adhesion molecule; PI3K, phosphoinositide 3-kinase; ROS, reactive oxygen species; SIRT, sirtuin; TET, ten-eleven translocation; VE, vascular endothelial; VEGFR, vascular endothelial growth factor receptor. Figure created using BioRender.

**Figure 2 cells-14-00378-f002:**
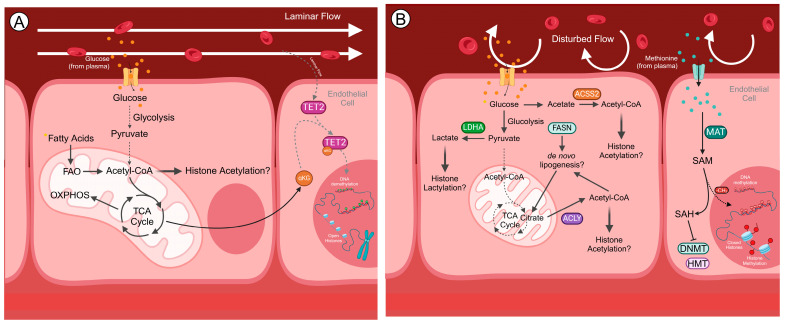
The relationship between metabolism and the EC epigenome under laminar and oscillatory shear stress. (**A**) ECs localized in areas where stable (atheroprotective) flow is prevalent are characterized by an upregulation of KLF2, which acts to suppress glucose transport and glycolysis. Mitochondria in healthy cells are mostly tubular, which is suggestive of a relatively high engagement in mitochondrial metabolism. Fatty acid oxidation plays a major role in maintaining the intracellular pool of acetyl-CoA, which is possibly used in histone acetylation reactions to preserve EC identity. Laminar flow is also characterized by the expression of TET2, which possibly acts to demethylate atheroprotective genes, using TCA cycle-derived α-KG as a cofactor. (**B**) By contrast, in ECs localized in regions of disturbed flow (atheroprone), aerobic glycolysis predominates, which is consistent with the fragmentation of the mitochondrial network. The increase in glycolysis is likely to generate lactate, which can be used in histone lactylation reactions. Acetate- and de novo lipogenesis-derived acetyl-CoA may also be used in histone acetylation. The hypermethylation observed in dysfunctional ECs is possibly attributed to the altered methionine uptake from plasma and/or altered one-carbon metabolism. Methionine is converted into SAM by MAT. SAM subsequently donates its methyl group to be used in DNA and histone methylation reactions. Abbreviations: ACLY, ATP citrate lyase; ACSS, acetyl-CoA synthetase; α-KG, alpha-ketoglutarate; CoA, coenzyme A; DNMT, DNA methyltransferase; FAO, fatty acid oxidation; FASN, fatty acid synthase; HMT, histone methyltransferase; LDH, lactate dehydrogenase; MAT, S-adenosylmethionine synthetase; OXPHOS, oxidative phosphorylation; SAM, S-adenosylmethionine; SAH, S-adenosylhomocysteine; TCA, tricarboxylic acid; TET, ten-eleven translocation. Figure created using BioRender.

**Table 1 cells-14-00378-t001:** Shear stress-sensitive genes in ECs. Abbreviations: EC, endothelial cell; EndMT, endothelial–mesenchymal transition; H_2_O_2_, hydrogen peroxide; LDL, low-density lipoprotein; NF-κB, nuclear factor kappa-light-chain-enhancer of B cells; NO, nitric oxide; ↑, increased expression; ↓, decreased expression.

Shear Stress	Gene	Expression	Effect in ECs	References
Laminar	*BMPR2*	↑	Inhibits NF-κB activation and oxidative stress	[54]
	*KDM4B*	↓	Induces EndMT	[55]
	*KLF2*	↑	Maintains EC identity	[27,56]
	*KLF4*	↑		[30,57]
	*NFE2L2*	↑	Reduces oxidative stress	[58]
	*NOS3*	↑	Catalyzes production of NO	[59]
	*PPARGC1A*	↑	Promotes mitochondrial biogenesis	[60]
	*PPAP2B*	↑	Reduces inflammation	[61]
	*SOD2*	↑	Antioxidative responses	[62]
	*SOD3*	↑		
	*SOX13*	↑	Reduces inflammation	[63]
	*TIMP3*	↑	Reduces extracellular matrix degradation	[64]
Oscillatory	*ACVRL1*	↓	Regulates angiogenesis, LDL entry and transcytosis	[24,65]
	*ADAMTSL5*	↓	Not established	[24]
	*BMP4*	↑	Enhances oxidative stress and inflammation	[66]
	*CCL2*	↑	Enhances immune cell adhesion to ECs	[67]
	*CMKLR1*	↓	Regulates ligand-mediated migration and angiogenesis	[24,68]
	*DOK4*	↓	Regulates NF-κB activation	[24,69]
	*F2RL1*	↓	Regulates angiogenic responses	[24]
	*GADD45*	↑	Activates cell growth and proliferation	[70]
	*HAND2*	↑	Increases extracellular matrix degradation	[71]
	*HIF1A*	↑	Enhances aerobic glycolysis	[72]
	*HOX5*	↓	Regulates angiogenesis	[24,73]
	*ICAM1*	↑	Enhances immune cell adhesion to ECs	[74]
	*KLF3*	↓	Regulation of NF-κB-driven inflammation	[24,75]
	*MAPK1*	↑	Activate cell growth and proliferation	[76]
	*MAPK3*	↑		
	*NFKB*	↑	Enhances inflammation	[77,78]
	*NOX1*	↑	Promotes production of superoxide	[79,80,81]
	*NOX2*	↑		
	*NOX4*	↑/↓	Promotes production of H_2_O_2_	[82,83,84]
	*PKP4*	↓	Regulates intercellular adhesion	[24,85]
	*SEMA7A*	↑	Enhances immune cell adhesion to ECs	[86]
	*SPRY2*	↓	Regulates integrity of endothelial cell monolayer	[24,87]
	*TMEM184B*	↓	Not established	[24]
	*TXNDC5*	↑	Impairs eNOS	[88]
	*VCAM1*	↑	Enhances immune cell adhesion to ECs	[74]
	*YAP1, TAZ*	↑	Enhances inflammation	[89]
	*ZFP46*	↓	Not established	[24]

## Data Availability

No new data were created or analyzed in this study.

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
