# Peer review of "Metaboloepigenetics: Role in the Regulation of Flow-Mediated Endothelial (Dys)Function and Atherosclerosis"

_cells, 2025, doi:10.3390/cells14050378_

Round 1
Reviewer 1 Report
Comments and Suggestions for Authors
This review is titled "Regulation of flow-mediated endothelial (dys)function: a role for metabolic epigenetics", which mainly discusses how flow regulates EC function through metaboloepigenetics mechanisms. This article is a very interesting review that systematically summarizes the role of shear force, metaboloepigenetics and EC dysfunction. This article has primary academic value and provides an important theoretical framework and future direction for research in this field.
Minor comments:
Please make the title more precisely to the MS.
Please define ‘metaboloepigenetics’ in the introduction.
Please check the abbreviations in the MS:
e.g. Page 12, line 453, ‘reactive oxygen species (ROS)’ should be changed to ‘ROS’, which has been defined in line183
Line 77: post-translational modifications (PTMs) again in line 179: post-translational modifications (PTMs)
Author Response
This review is titled "Regulation of flow-mediated endothelial (dys)function: a role for
metabolic epigenetics", which mainly discusses how flow regulates EC function through
metaboloepigenetics mechanisms. This article is a very interesting review that
systematically summarizes the role of shear force, metaboloepigenetics and EC
dysfunction. This article has primary academic value and provides an important theoretical
framework and future direction for research in this field.
We are grateful for the supportive comments of this reviewer
Minor comments:
Please make the title more precisely to the MS.
We have changed the title to “A role for metaboloepigenetics in the regulation of flow-mediated endothelial (dys)function and atherosclerosis” to better reflect the content of the manuscript.
Please define ‘metaboloepigenetics’ in the introduction.
This has now been defined in the introduction (row 70).
Please check the abbreviations in the MS:
e.g. Page 12, line 453, ‘reactive oxygen species (ROS)’ should be changed to ‘ROS’,
which has been defined in line183
This has now been changed. We have also defined all abbreviations at the end of the manuscript
Line 77: post-translational modifications (PTMs) again in line 179: post-translational
modifications (PTMs).
These abbreviations have now been put in place.
Reviewer 2 Report
Comments and Suggestions for Authors
At the begining, I would like to thank you for choosing me to do this review.
The article entitled "Regulation of flow-mediated endothelial (dys)function: a role 1 for metaboloepigenetics" presents a very discussed topic at the moment. Unfortunately, however, there are several aspects that need to be modified.
First of all, the article does not respect the sections recommended for writing all manuscripts. I ask the authors to consult the journal's guide for authors. The body of the text requires reorganization according to these considerations.
The introduction is well organized and highlights the current state of knowledge regarding the topic discussed.
Sections 2, 3, and 4 need to be reorganized.
Second, to the material and methods section, which does not exist now, the working methodology that guided the authors in this manuscript should be added, where at least part of it is a review.
Section 5 should be added, partially to the results. This manuscript contains almost no discussion of the results found.
Third, I ask the authors to mention the type of article proposed for publication. Thus, if it is a review article, to mention whether it is a perspective, scoping or systematic one and, possibly, to use the general guidelines in this regard.
Captions for figures and table must be included. A list of abbreviations must also be included.
Also, the references section contains some old and very old bibliographic indexes. If possible, these should be updated. References must be numbered in order of appearance in the text and numbers should be placed in square brackets [ ], and placed before the punctuation.
Author Response
At the begining, I would like to thank you for choosing me to do this review.
The article entitled "Regulation of flow-mediated endothelial (dys)function: a role 1
for metaboloepigenetics" presents a very discussed topic at the moment. Unfortunately,
however, there are several aspects that need to be modified.
First of all, the article does not respect the sections recommended for writing all
manuscripts. I ask the authors to consult the journal's guide for authors. The body of the
text requires reorganization according to these considerations.
This issue has been addressed as detailed below.
The introduction is well organized and highlights the current state of knowledge
regarding the topic discussed.
Sections 2, 3, and 4 need to be reorganized.
We requested guidance from the editor on our organisation, and we directed thus “Review sections: a literature review organized logically within specific sections and subsections (optional).”
We believe that our manuscript conforms with this.
Second, to the material and methods section, which does not exist now, the working
methodology that guided the authors in this manuscript should be added, where at least part
of it is a review.
We have been advised by the editors that a methods section is not appropriate for a review article
Section 5 should be added, partially to the results. This manuscript contains almost
no discussion of the results found.
As this is a review article, the data described within the manuscript are discussed as they are introduced. Therefore a discussion of the data at the end is not appropriate here.
Third, I ask the authors to mention the type of article proposed for publication.
Thus, if it is a review article, to mention whether it is a perspective, scoping or systematic
one and, possibly, to use the general guidelines in this regard.
This is a review article. This information has been included at the beginning of the article and in the information accompanying the resubmitted manuscript.
Captions for figures and table must be included. A list of abbreviations must also be
included.
Captions for figures and tables have now been included and are marked in bold.
Also, the references section contains some old and very old bibliographic indexes. If
possible, these should be updated. References must be numbered in order of appearance in
the text and numbers should be placed in square brackets [ ], and placed before the
punctuation.
Some of the older references are seminal papers which we feel it is necessary to include. The references are now cited in the style of the journal.
Reviewer 3 Report
Comments and Suggestions for Authors
Main Research Question: The article seeks to answer the question of how mechanotransduction and epigenetic changes regulate endothelial function and how cellular metabolism may influence these processes, leading to the development of atherosclerosis. The authors focus on the role of metabolic epigenetics in endothelial pathology, which is a relatively novel approach in the context of cardiovascular diseases.
Originality and Relevance of the Topic: The topic of the article is original and relevant to the field of atherosclerosis research. Although the relationship between mechanotransduction and endothelial function has been previously studied, the authors introduce a new perspective by linking these mechanisms to metabolic epigenetics. This approach may help fill a gap in understanding how metabolic changes influence endothelial epigenetics and how this may lead to endothelial dysfunction and atherosclerosis.
Contribution to the Field: The article makes a significant contribution to the field by integrating existing research on mechanotransduction, epigenetics, and metabolism in the context of endothelial pathology. Unlike other publications that often focus on one of these aspects, the authors propose a holistic approach that may open new research directions for the treatment of atherosclerosis.
Comments on Methodology: The article is a review, so it does not contain new experimental data. However, the authors could consider discussing the limitations of current research methods.
Conclusions: The conclusions are generally consistent with the evidence and arguments presented. The authors effectively combine various aspects of research to suggest that metabolic changes may influence endothelial epigenetics, which in turn may lead to atherosclerosis. However, some hypotheses, such as the role of specific metabolites (e.g., acetyl-CoA or α-ketoglutarate) in modulating the epigenome, require further research to be confirmed.
References: The article is well-documented, and the authors refer to the latest research in the field.
Comments on Tables and Figures: Table 1: Abbreviations in the table should be explained in the legend to facilitate reader understanding. Additionally, it would be worth considering adding a column explaining how changes in gene expression affect endothelial function.
Additional Comments: What is the purpose of the question mark in the title of paragraph 5 (line 390)?
Author Response
Main Research Question: The article seeks to answer the question of how
mechanotransduction and epigenetic changes regulate endothelial function and how cellular
metabolism may influence these processes, leading to the development of atherosclerosis.
The authors focus on the role of metabolic epigenetics in endothelial pathology, which is a
relatively novel approach in the context of cardiovascular diseases.
Originality and Relevance of the Topic: The topic of the article is original and relevant to
the field of atherosclerosis research. Although the relationship between
mechanotransduction and endothelial function has been previously studied, the authors
introduce a new perspective by linking these mechanisms to metabolic epigenetics. This
approach may help fill a gap in understanding how metabolic changes influence endothelial
epigenetics and how this may lead to endothelial dysfunction and atherosclerosis.
Contribution to the Field: The article makes a significant contribution to the field by
integrating existing research on mechanotransduction, epigenetics, and metabolism in the
context of endothelial pathology. Unlike other publications that often focus on one of these
aspects, the authors propose a holistic approach that may open new research directions for
the treatment of atherosclerosis.
We would like to thank this reviewer for their supportive comments.
Comments on Methodology: The article is a review, so it does not contain new
experimental data. However, the authors could consider discussing the limitations of
current research methods.
The research limitations are now discussed in the conclusions (lines 584-586)
Conclusions: The conclusions are generally consistent with the evidence and arguments
presented. The authors effectively combine various aspects of research to suggest that
metabolic changes may influence endothelial epigenetics, which in turn may lead to
atherosclerosis. However, some hypotheses, such as the role of specific metabolites (e.g.,
acetyl-CoA or α-ketoglutarate) in modulating the epigenome, require further research to be
confirmed.
This point has now been made in the conclusion section (rows 606-608)
References: The article is well-documented, and the authors refer to the latest research in
the field.
Comments on Tables and Figures: Table 1: Abbreviations in the table should be
explained in the legend to facilitate reader understanding.
Definitions have now been included in all the figure and table legends.
Additionally, it would be worth
considering adding a column explaining how changes in gene expression affect endothelial
function.
Unfortunately the functional effects of the changes in gene expression are not always known. We therefore include in the text the potential functional impact of altered expression, as appropriate. However, are not able to give a table documenting all changes.
Additional Comments: What is the purpose of the question mark in the title of paragraph
5 (line 390)?
This has now been removed.